# Heterologous Interactions with Galectins and Chemokines and Their Functional Consequences

**DOI:** 10.3390/ijms241814083

**Published:** 2023-09-14

**Authors:** Kevin H. Mayo

**Affiliations:** Department of Biochemistry, Molecular Biology & Biophysics, University of Minnesota Health Sciences Center, 6-155 Jackson Hall, Minneapolis, MN 55455, USA; mayox001@umn.edu

**Keywords:** galectin, chemokine, structure, function, NMR, modeling

## Abstract

Extra- and intra-cellular activity occurs under the direction of numerous inter-molecular interactions, and in any tissue or cell, molecules are densely packed, thus promoting those molecular interactions. Galectins and chemokines, the focus of this review, are small, protein effector molecules that mediate various cellular functions—in particular, cell adhesion and migration—as well as cell signaling/activation. In the past, researchers have reported that combinations of these (and other) effector molecules act separately, yet sometimes in concert, but nevertheless physically apart and via their individual cell receptors. This view that each effector molecule functions independently of the other limits our thinking about functional versatility and cooperation, and, in turn, ignores the prospect of physiologically important inter-molecular interactions, especially when both molecules are present or co-expressed in the same cellular environment. This review is focused on such protein-protein interactions with chemokines and galectins, the homo- and hetero-oligomeric structures that they can form, and the functional consequences of those paired interactions.

## 1. Introduction

Galectins and chemokines have been known, essentially since their discovery, to form homodimers, as well as some higher order homo-oligomers. However, at the turn of the millennium, two CC chemokines (macrophage inflammatory proteins 1α and 1β) were reported to be secreted simultaneously from cells as heterodimers [1]. At that time, those authors thought it was a unique phenomenon with chemokines, and the observation was reported without experimental validation of its functional significance [1]. However, shortly thereafter it was discovered that multiple chemokines indeed cross-react inter-molecularly as CXC-CXC and CC-CC chemokines, as well as CC-CXC paired chemokines, and with functional significance [2,3,4]. This was quickly followed by further experimental confirmation of the functional significance of paired chemokines [5,6]. In these instances, NMR and molecular modeling was used to demonstrate their physical interactions as heterodimers that modulate their respective functions. The next observation was that galectins too could cross-interact with each other as galectin-galectin heterodimers [7,8]. Finally, because some chemokines and galectins were found to be present in the same cellular micro-environment, this concept was extended to show that members from each family could interact as galectin-chemokine heterodimers with functional consequences [9]. Since these initial reports, a number of other research labs have picked up on this concept and have started to investigate such inter-molecular interactions in other systems.

Due to growing interest in this area, this review is focused on chemokines and galectins, their structures and interactions, and consequent functional effects. We first present some general background on chemokines and galectins, followed by experimental validation of their structural interactions as homo- and hetero-dimers (both within each family and then inter-family as chemokine–galectin heterodimers), and finally the functional consequences thereof.

## 2. Chemokines

Chemokines are a family of small, highly conserved proteins (~8 kDa to ~12 kDa) involved in many biological processes, including chemotaxis [10], leukocyte degranulation [11], hematopoiesis [12], and angiogenesis [13,14]. Chemokines function by interacting with cell surface G-protein coupled receptors (GPCRs), and are usually categorized into sub-families that are based on the sequential positioning of the first two of four highly conserved cysteine residues: CXC, CC, and CX3C [15]. The C chemokine sub-family is the exception with only one N-terminal cysteine residue. In the largest subfamilies, CC and CXC, the first two cysteines are adjacent (CC motif) or separated by one amino acid residue (CXC motif). C-type chemokines lack the first and third of these cystines, and CX3C chemokines have three amino acids between the first two cysteine residues. Even though sequence identity between chemokines varies from about 20% to 90%, their sequences overall are highly conserved, and all chemokines adopt the same monomer fold that consists of a flexible N-terminus and loop, followed by a three-stranded antiparallel β-sheet on to which is folded a C-terminal α-helix, e.g., [16]. The highly conserved cysteine residues (four in CXC and CC chemokines) pair up to form disulfide bridges that are crucial to maintaining structural integrity and binding to their respective GPCRs [17].

Chemokine monomers usually associate to form oligomers, primarily dimers, but some are also known to form tetramers [18,19] and higher order species, e.g., [20,21]. Despite their highly conserved monomer structures, chemokines form different types of homodimer structures, depending upon which sub-family they are associated [16]. In general, homodimer structures are determined primarily by the composition of amino acid residues that are present at the particular inter-subunit interface [4]. Within each chemokine sub-family (i.e., CC or CXC), homodimer structures are essentially the same, albeit quite different from those in another family. Figure 1A,B shows homodimer structures of CXC chemokine CXCL8 (Interleukin-8) [22] and CC chemokine CCL5 (RANTES) [23]. The more globular CXC-type dimer (Figure 1A) is formed by interactions between β1 strands from each monomer subunit that extend the three stranded anti-parallel β-sheet from each monomer into a six-stranded β-sheet, on top of which are folded the two C-terminal α-helices that run in an antiparallel fashion. On the other hand, CC-type chemokines (Figure 1B) form elongated end-to end-type dimers through contacts between their short N-terminal β-strands (labeled βN) with the two C-terminal helices running almost perpendicular to each other on opposite sides of the molecule. Nevertheless, some CC-type dimer structures such as CCL5 have been reported to differ in the relative orientation of some secondary structure elements (e.g., C-terminal α-helices), although this may be related to differences in structural dynamics and/or crystal lattice effects [21,24]. Chemokines can also oligomerize to tetramers with CXCL4 (platelet factor-4) [25], where two CXC-type dimers associate to form a β-sandwich with the β-sheet of one dimer lying on top of the β-sheet of the other. Tetramers have also been observed with CXCL7 [18], and their higher order structures can vary considerably, e.g., CXCL10 [26] that forms both CC- and CXC-type homodimer topologies. Some CC-type chemokines even associate as higher order oligomers, e.g., CCL5 [20] and CCL27 [21].

Chemokine monomers, homodimers and higher order homo-oligomers exist in a complex equilibrium where distinct oligomer structures co-exist and interconvert within a dynamic distribution [20,21,27,28,29,30,31]. This is exemplified by NMR studies on CXCL4 [32,33], low affinity CXCL4 [34], and CXCL7 (platelet basic protein) [18] and its N-terminal degradation products CTAP-III (connective tissue-activating peptide III) and NAP-2 (neutrophil-activating peptide-2) [35]. The weighting factor with oligomer populations is dictated by thermodynamic stability and the amino acid composition/conformation of their inter-subunit interfaces [4]. Some chemokines form strong oligomers and others form much weaker ones or remain as monomers/dimers. This equilibrium can be perturbed by changing solution conditions (e.g., lower pH, buffer type, ionic strength, presence of ligands such as heparin), e.g., with CXCL4 [32,33,36], CXCL7 [37], CXCL12 [38,39], and CCL11 (exotoxin) [40]). The oligomer subunit exchange rate provides a reasonable explanation as to why some chemokines cannot be crystallized or why their structures cannot be solved by NMR.

Chemokines are also known to bind specifically and strongly to GAGs (glycosaminoglycans) [30,41,42,43]. CCL5 homodimers bind GAGs [44,45,46] with binding affinities depending on the type of GAG and its sulfation pattern [46]. Similar observations have been made for CCL2 [47] and CCL11 [41]. Overall, the highly negatively charged GAGs interact electrostatically with positively charged residues in these chemokines. Contrary to some CXCL4/GAG binding models that focus on clusters of lysines in the chemokine C-terminal α-helix, Mayo et al. [48] used NMR and site-directed mutagenesis to demonstrate that the loop containing Arg20, Arg22, His23 and Thr25, as well as Lys46 and Arg49, also plays a significant role in GAG/heparin binding. GAG binding can even induce chemokine oligomerization [49], as exemplified with CCL5 [30,50,51]. GAGs can also affect chemokine structure [30,47], structural dynamics [44], and chemokine receptor dimerization, e.g., CCR2 [52]. For example, Rek et al. [30] found that there is a structural transition in CCL5 upon GAG binding, and Verkaar et al. [53] discovered that GAGs affect chemokine cooperativity. Furthermore, Mikhailov et al. [54] showed early on that heparin dodecasaccharide binding to CXCL4 induces higher order oligomer formation that is dependent upon the chemokine-to-GAG molar ratio.

### 2.1. Chemokine Heterodimers

Because chemokine monomer structures are highly conserved (see Figure 1A), heterodimer structures, similarly to homodimer structures, are determined primarily by amino acid residues at the particular inter-subunit interface of the interacting chemokine monomers [4]. Therefore, monomers of different chemokines may be swapped if the arrangement and composition of residues at a given monomer-monomer interface in the heterodimer make for a more energetically favorable state relative to that in either homodimer. Modeled structures of CXCL4 and CXCL8 homodimers, as well as of the CXCL4/CXCL8 heterodimer, are shown in Figure 2A–C. These structures were guided by NMR HSQC chemical shift and intensity changes from ^15^N-labeled CXCL4 and CCL5, followed by manual docking and energy minimization using MD simulations. Positively charged lysine and arginine residues (shown in dark blue) electrostatically interact with negatively charged glutamate residues (shown in dark red) to thermodynamically stabilize both homo- and hetero-dimers.

Guan et al. [1] showed that CC chemokines CCL3/4 and CCL2/8 (macrophage inflammatory protein 1β (MIP-1β) and 1α (MIP-1α), respectively) are secreted together from activated human monocytes and peripheral blood lymphocytes, thereby forming heterodimers in vitro, yet only speculating (but not experimentally proving) that this heterodimer pair effects intracellular signaling by binding to, and activating, its CCR5 cell receptor [1]. However, we now know that even chemokines that are not simultaneously secreted from cells do form heterodimers. For example, at least three members of the CXC sub-family: CXCL4 [3,32], its N-terminal chimera PF4-M2 [25], and CXCL8 [3], were shown early on to exchange subunits and form heterodimers with similar equilibrium dissociation constants (K_D_) as observed for homodimers [2,3]. Moreover, CXCL4/CXCL8 heterodimers were shown to be functionally significant in vitro in cell-based assays [3].

Nesmelova et al. [4] investigated the energetic basis for CC and CXC heterodimer formation by using molecular mechanics and the Poisson-Boltzmann surface area (MM-PBSA) approach to calculate binding free energies and to predict which pairs of CXC and CC chemokines would likely form in solution. This study indicated that heterodimers within and between members of CXC and CC sub-families can occur. Calculations were performed to assess also which type of heterodimer might form, i.e., CXC-type vs. CC-type heterodimers. In this regard, CXCL4 can make thermodynamically favorable interactions with CXCL1, CXCL7, and CXCL8, as well as CXCL1/L8, CXCL7/L8 and CXCL1/L7, with all hetero-pairs forming only CXC-type dimers. CC chemokine CCL2 also favorably paired with CCL5 and CCL8, with CXC-type heterodimers being favored with the CCL2/CCL8 pair. Several CXC/CC mixed chemokine pairs were also examined, with CCL2/CXCL4 and CCL2/CXCL8 favoring CXC-type heterodimers, and CCL5/CXCL4 greatly favoring CC-type heterodimers, and CXCL8/CCL5 forming both equally well.

Von Hundelshausen et al. [6] used this in silico approach [4] with CXCL4 and CCL5, in which CXCL4 and CCL5 monomers were initially docked as both CXC- and CC-type heterodimers, with the CC-type heterodimer being more energetically favored. Moreover, the formation of the CXCL4/CCL5 heterodimer was validated experimentally both in vitro and in vivo [5,6]. More recently, Brandhofer et al. [55] demonstrated that heterocomplexes also form between atypical chemokine MIF (macrophage migration inhibitory factor) and CXCL4L1 and regulate inflammation and thrombus formation in patients with peripheral artery disease.

Chemokine heterodimers, similarly to their homodimer counterparts, are also stabilized by binding to GAGs [56], as exemplified with CCR2 ligands CCL2 (MCP-1), CCL8 (MCP-2), CCL7 (MCP-3), CCL13 (MCP-4), and CCL11 (eotaxin) [52]. In particular, CCL2 and CCL8 form strong CC-type heterodimers, whereas CCL2/CCL13, CCL2/CCL11 and CCL8/CCL13 heterodimers are only moderately stable, and CCL7 does not form heterodimers with any other CCR2 chemokine ligand. In addition, the heparin pentasaccharide Arixtra promotes formation of CCL8/CCL11 and CCL2/CCL11 heterodimers, which otherwise do not form or form only weakly.

### 2.2. Functional Considerations

Chemokines play a significant role in biology and are involved in many pathologic disorders, including cancer, HIV/AIDS, and atherosclerosis [57,58,59]. About fifty chemokines are involved in various aspects of cell interactions and communication within the immune system. In general, chemokines trigger their functional activities by binding to cell surface G-protein coupled receptors (GPCRs) [60,61,62,63]. Determining high resolution structures of chemokine ligand/receptor complexes is highly challenging, and few structures are presently known. However, Handel et al., for example, have reported on a complex formed between chemokine MCP1 and its receptor CCR2 [64], and more recently, on an X-ray crystal structure of chemokine vMIPII and its receptor CXCR4 showing that it is the vMIPII monomer, and not the homodimer, that binds to CXCR4 [65]. In studies such as these, structural biology plays a major role in delineating how chemokine ligands interact with their GPCR receptors, which in turn relates to how chemokines trigger cell signaling. Because GPCRs are difficult to purify and work with, studies have generally focused on identifying residues in the chemokine ligands responsible for activating GPCRs. For example, the Glu-Leu-Arg tripeptide at the N-terminus of some CXC chemokines is crucial for interactions with GPCRs. CXC chemokines with the ELR motif (ELR1: CXCL1, 2, 3, 5, 6, 7, 8) promote angiogenesis and those lacking the ELR motif (ELR2: CXCL4, 9, 10) are reported to be angiostatic [14,66]. Residues within the N-loop between the two N-terminal cysteines, as well as in the helix, can also be involved in GPCR binding [19,67]. Binding interactions vary from chemokine to chemokine and the specific GPCR.

Chemokine heterodimers also mediate some cellular responses [1,2,3,4,5,6,28,68,69,70]. For example, CXCL8/CXCL4 heterodimers enhance CXCL4-induced endothelial cell proliferation and CXCL8-induced migration of Baf3 cells [3], whereas the co-presence of CXCL4 and CXCL8 attenuates a CXCL8-mediated rise in intracellular calcium in the amyeloid progenitor cell line and enhances CXCL8-induced migration of pro-B-cells (Baf/3) [2,3]. Heterodimerization between members of the CXC and CC sub-families has been reported with CXCL4 and CCL5 [5,6,70], as well as with CCL21 (secondary lymphoid tissue chemokine) and CXCL13 (B cell attracting chemokine-1) [28]. The functional result is that hetero-dimerization dramatically modulates chemokine-mediated activities in vitro and in vivo. Chemokine heterodimerization can affect the overall signaling response of GPCRs, thereby providing a general mechanism for regulating chemokine function. The recent synthesis and in vitro/in vivo testing of a covalently-linked CXCL4/CCL5 heterodimer has validated the functional relevance of chemokine heterodimers in GPCR-mediated signal transduction [6,71]. Another covalently-linked heterodimer (termed OHD 4–12) was produced by Nguyen et al. [72] between CXCL4 and CXCL12. OHD 4–12 was shown to inhibit CXCL12-driven migration of triple-negative MDA-MB-231 breast cancer cells, and to bind to and activate CXCR4 receptors [72].

As mentioned above, GAGs are essential to chemokine function in vivo [51,73,74]. In fact, GAGs also affect chemokine heterodimer function, as exemplified with chondroitin sulfate that appears to induce CCL5/CXCL4 heterodimers and promote atherosclerosis [5,70]. The possibility of functionally relevant, structurally distinct oligomer conformations as a result of GAG/chemokine interactions has been postulated [44,75]. Mikhailov et al. [54] showed that a heparin-based dodecasaccharide binding to CXCL4 induces the formation of higher order oligomers which depends on the molar ratio of chemokine:GAG, and thereby can lead to the pathological disorder of thrombocytopenia. Even though there is evidence that chemokine heterodimers impact biological activity, this does not exclude the occurrence of individual chemokines working in concert on their respective GPCRs to elicit synergistic effects. In fact, Gouwy et al. [76,77,78] demonstrated that blocking one of two chemokine receptors negates synergistic interactions, suggesting that synergy requires each chemokine to bind to its respective cell receptor leading to intra-cellular signaling. It is likely that there are several ways to obtain synergy with these chemo-attractants [79].

Whereas initial reports of chemokine heterodimers [1,2,3,4] were viewed as somewhat controversial in terms of their biological relevance, the heterodimer concept has been validated experimentally and presents a novel paradigm for designing chemokine antagonists [3,5,6]. For example, Koenen et al. [5] reported on the use of CCL5-derived peptides, e.g., CKEY, that function as chemokine heterodimer agonists. The term “chemokine interactome” was recently introduced to promote the chemokine heterodimer concept and present further empirical evidence as to which CXC and CC chemokines interact physically with each other [6]. This large sampling of chemokines demonstrated that not all of them can associate as heterodimers, which imparts selectivity to interactions between mixed chemokines. Moreover, the GAG chondroitin sulfate appears to induce a specific CCL5/CXCL4 heterodimer that enhances atherosclerotic development [5,70]. Other chemokine-derived peptides are also effective chemokine antagonists [6]. However, it was the design and synthesis of two covalently-linked CXCL4/CCL5 [71] and CXCL4/CXCL12 heterodimers [72] that has provided the most compelling evidence in vitro and in vivo that validates the biological relevance of chemokine heterodimers.

## 3. Galectins

Aside from chemokines, another class of effector molecules that stands out from other regulators is the family of galectins [80,81]. As a sub-family of lectins, galectins have a high level of sequence similarity in their carbohydrate recognition domains (CRDs) along with a preference (albeit non-exclusive) for binding β-galactosides [80]. Even though there are ~14–16 known galectins [82,83,84], galectin-1 (Gal-1) is by far the most studied [85], primarily because it was the first one reported [86]. Galectins are separated into three groups: prototype (Gal-1, 2, 5, 7, 10, 11, 13, 14), chimera (Gal-3), and tandem repeat (Gal-4, 6, 8, 9, 12) [87,88]. Whereas prototype galectins have one CRD, tandem repeat galectins have two homologous CRDs linked by a flexible peptide linker. Gal-3 is the only chimera-type galectin with its conserved CRD tethered to a very long, dynamic, proline-rich N-terminal tail (NT) comprised of multiple PGAY-tetrapeptide repeats. Transient interactions between the Gal-3 CRD and its NT [89] are reported to have a significant effect on glycan binding and various Gal-3-mediated cell-based activities [90]. Moreover, Zhao et al. [90] demonstrated that mutation of any single Gal-3 NT proline (especially P37, P55, P60, P64, and P67) dramatically and differentially inhibits Gal-3–mediated cellular activities (e.g., cell migration, activation, endocytosis, and hemagglutination), as well as glycoprotein-binding-induced Gal-3-mediated oligomerization (via liquid–liquid phase separation, LLPS), a fundamental process required for various activities. Some of the more recently identified galectins show some unusual behavior. For example, recombinant Gal-16 has a pseudo ligand binding site that plays a role in regulating c-Rel-mediated lymphocyte activity [91]; recombinant Gal-14 stands out as an important effector molecule in embryonic development [92], and actin binding to recombinant Gal-13 occurs independently of the galectin canonical ligand binding site [93]. In this regard, Gal-13, -14 and -16 are highly conserved compared to other galectins, are mostly expressed in the placenta of anthropoids (particularly at the maternal-fetal interface), and do not readily bind lactose unless specific residues within the canonical sugar binding site are mutated, e.g., Gal-13 R53 and H57 changed to H53 and R57, and Gal-16 R55 substituted as N55 [91,92,93].

The CRD structures of numerous galectins, usually bound to the disaccharide lactose, have been reported (search the Protein Data Bank, PDB). All galectin CRDs fold as β-sandwiches formed from two β-sheets, one with six anti-parallel β-strands (the sugar binding S-face) and the other with five anti-parallel β-strands (the opposing F-face) (see Figure 3). Their conserved folded structures result from their amino acid sequences being highly conserved [94], another commonality shared with chemokines. Gal-1 is unique, because it has six cysteine residues that need to be oxidized into three disulfide bridges for the lectin to be active in binding carbohydrates [95,96]. There are, however, no known structures of wild type tandem repeat galectins due to the presence of their flexible linker or of full length Gal-3 due to the presence of its long, dynamic N-terminal tail.

Galectins normally bind to β-galactosides or β-galactoside-containing glycans on the cell surface or within the extracellular matrix. Binding occurs at a site in their CRD that is formed by highly conserved residues in the six-stranded β-sheet S-face (Figure 3B) [94]. Upon binding the β-galactoside lactose, for example, CRD residues within the carbohydrate binding site generally do not vary much in terms of their conformations. The major change occurs with the orientation of the highly conserved tryptophan ring that is disordered in the ligand-free state and highly ordered in the lactose bound state as it interacts with the galactose ring. Although the conformation may not change much, internal motions or dynamics can, as evidenced with Gal-1 where lactose binding modulates structural dynamics, increases conformational entropy, and occurs with apparent negative cooperativity [97]. Lactose binding to Gal-7 also induces long-range, allosteric effects through the protein resulting in increased homodimer stability and positive ligand binding cooperativity [98]. With Gal-7, proline isomerization impacts its structural dynamics and glycan affinity [99].

Galectins bind to many carbohydrate ligands, e.g., *N*-acetyllactosamine octasac-charide [100], *N*-acetyllactosamine disaccharide [101] and GalNAc-Man-related glycans [102]. In each instance, the non-reducing end of the sugar binds in a similar way, in particular via its interaction with the conserved tryptophan mentioned above. However, with N-acetyllactosamine octasaccharide, the remainder of its longer oligosaccharide chain points out into solution, thus possibly creating a scenario in which a second galectin dimer binds to the free end of the octamer to promote extended polymerization, something that has yet to experimentally verified.

Although β-galactosides are the generally-accepted recognition elements to which galectins bind, this is not always the case, because α-linked digalactosides for example have also been shown to bind to Gal-1, albeit more weakly [103]. Moreover, natural galectin-binding glycans on the cell surface are much more complicated and may determine and/or differentiate galectin function. For example, Gal-3 binds to endothelial cell adhesion molecule CD146 in a somewhat non-conventional way with non-canonical interactions between glycans on CD146 and the lectin’s CRD β-sandwich F-face [104], and Gal-3 binds selectively to the terminal non-reducing end of β(1–4)-galactans with affinity increasing with chain length. Moreover, polysaccharides with a larger glycan footprint bind Gal-1 in a more extensive fashion through most of the CRD S-face [105], and an α-galactomannan (i.e., Davanat) binds Gal-1 at a non-canonical site on its CRD [106,107]. Furthermore, β(1–4)-polymannan and α-branched mannan also bind to Gal-3 at a non-canonical site on the F-face of its CRD, and apparently to a lesser extent to the canonical carbohydrate binding site on the CRD S-face [108]. Polysaccharides may also interact with the N-terminal tail (NT) of Gal-3, a process that is likely modulated by NT proline isomerization [109], and macromolecular assemblies of complex polysaccharides can interact with Gal-3 with synergestic effects on function [110]. In addition, Zheng et al. [111] showed that negatively-charged homogalacturonan-derived oligosaccharides bind to the Gal-3 CRD canonical site in a “topsy-turvy” fashion with their reducing end residue (as opposed to the non-reducing end residues as in lactose) interacting with the conserved tryptophan. This scenario may well be how galectins bind to heparan sulfate-based glycoconjugates on the surfaces of cells, as further exemplified by the binding of keratan sulfate and related saccharides to Gal-1, -3, -7 and -9N [112].

Polysaccharide branching also affects galectin binding. Gal-1 and -7 show greater binding affinities when the oligosaccharide is branched, whereas Gal-8 shows decreased affinity for branched oligosaccharides [113]. The variance of galectin binding to branched or non-branched glycans may provide for different galectins to modulate their biological functions, which can be quite different, E.g., Gal-1 induces T-cell apoptosis, whereas Gal-3 can suppress apoptosis and increase T-cell proliferation [114]. Thus, the function of any one galectin can be multi-faceted, and galectin dimerization and the more varied and complicated nature of glycans in situ, as well as interactions with other biomolecules, will play critical roles in differentiating galectin function.

Similarly to chemokines, galectin monomers can also associate as homodimers (especially prototype galectins), and in some cases, higher oligomer states. Regardless of their having similar monomer folds, galectins, similarly to chemokines, can form different types of homodimers via different surface interactions. Because the folded structures of all galectin monomers are essentially the same, the formation of any one type of homodimer structure is thermodynamically explained by the chemical composition of the set of residues at the monomer-monomer interface. Additionally, as with chemokines, when free energies/residue are in toto greater for one dimer type over another, then that one will dominate. For example, the Gal-1 homodimer is formed primarily via hydrophobic interactions between the N- and C-termini of two monomers (Figure 3A), whereas the Gal-7 homodimer is formed primarily by electrostatic interactions between residues from the two CRD β-sandwich F-faces (Figure 3B). A given homodimer type may be functionally important over another, and possibly explain why different galectins bind differently to complex glycans than to simple disaccharides (e.g., lactose) that are usually employed to investigate galectin-carbohydrate binding and function.

In solution, Gal-1 forms symmetric homodimers [97], whereas the crystal structure of Gal-1 shows the presence of asymmetric dimers, possibly the result of crystal lattice force-based distortion. In situ, it is more likely that symmetric Gal-1 homodimers form. In solution, Gal-3 appears either as a monomer [115] or homodimer [116], and although Ahmad et al. reported that Gal-3 precipitates from solution in a pentameric state (via interactions among their N-Terminal tails (NTs)) upon binding synthetic carbohydrates [117], this is generally unconfirmed in other studies. Nevertheless, the NT of Gal-3 has been reported to mediate higher order oligomerization [118], and recently, it was shown that Gal-3 in the presence of various glycoproteins (i.e., CD146, CD45, CD71, and CD7) oligomerizes through its NT (albeit not as a pentamer) to initiate liquid–liquid phase separation (LLPS), a fundamental process required for these cell activities [90].

The oligomerization state of a galectin can differ with varying solution conditions and thereby affect function, as exemplified with Gal-5 and -7 that behave as monomers [119,120], yet Gal-5 is active at hemagglutination. Gal-7 in crystals is a homodimer, whereas Gal-7 in solution is observed in a dynamic equilibrium between monomers and homodimers [121]. On the other hand, prototype Gal-10 spontaneously forms Charcot-Leyden crystals [122], and Gal-13 homodimers are linked via disulfide bonds. A natural form of Gal-1 (Gal-1β) that lacks the initial six residues, remains monomeric [123] and promotes axonal regeneration, but not Jurkat cell death, unlike Gal-1 homodimers that promote both [123]. In this instance, the homodimeric state is important to function, and changes to that state can be consequential to function. Even though the galectin oligomer state can be crucial to function, many papers often do not discuss it when explaining function.

Whereas some galectins have intracellular functions, they are usually most active extracellularly on the cell surface and within the extracellular matrix. In their oligomeric state, galectins normally function by cross-linking glycans on the cell surface [124] where they bind numerous glycoconjugates to trigger some function, e.g., cell-cell and cell-matrix interactions, proliferation, apoptosis, adhesion, migration, immunity, and inflammation [125]. Galectins are perhaps most generally associated with immune cells and cells in the cardiovascular system, and some galectins are associated with cell growth and differentiation. In this regard, galectins have been described as “exquisite modulators of the immune response” [84]. As broad-spectrum effectors of leukocyte activity and migration [126,127,128], galectins serve as bridges between cells and cell surface glycoconjugates to form lattices in membranes [129,130,131,132,133,134] and to act as sensors of damage- and pathogen-associated effects [135,136]. Cellular expression of galectins varies by cell type and activation state, with any given cell expressing at least one galectin [137]. For example, Gal-1 binds glycoconjugates in the extracellular matrix and on endothelial cells [138] and T lymphocytes [139]. Gal-3 also binds glycoconjugates in the extracellular matrix, as well as CD43 and CD45 on leukocytes, and can be induced by inflammatory mediators [140], e.g., CXC chemokine CXCL8 (interleukin-8, IL8) [141]. Neutrophils from peripheral blood are not responsive to Gal-3, yet neutrophils harvested from an inflammatory site are [142], suggesting glycan modifications are at play. In addition, while Gal-1-induces T-cell apoptosis, Gal-3 may suppress it and enhance T-cell proliferation [114]. Gal-3 is likely the most promiscuous lectin, because it displays diverse biological activities from cell adhesion, apoptosis, and immune regulation. Overall, galectins play various roles to control and mediate different biological functions.

### 3.1. Galectin-Galectin Heterodimers

Because galectins, similarly to chemokines, are small, protein effector molecules with highly conserved monomeric structures that form different types of homodimers, Miller et al. first proposed that galectins too could form heterodimers between/among themselves with functional significance [7]. As with chemokines, this is especially true when galectins are temporally and spatially co-expressed. The report of physical interactions between two C-type tissue lectins dectin-3 (MCL) and dectin-2 or Mincle [143,144] supports this concept. Using resin-bound Gal-7 in an affinity chromatography-based assay, Miller et al. [7] found that the CRDs of Gal-1 and Gal-3 bound to Gal-7, thus demonstrating hybrid formation between these paired galectins. Moreover, tandem-repeat Gal-8 also associated with Gal-7 on the resin, as did Gal-1 variants where either two CRDs were covalently linked by a Gly-Gly spacer or with the Gal-8 linker sequence (i.e., Gal-1 [GG] or Gal-1 [8S] [145,146]). Since secreted galectins bind cell surface receptors, Miller et al. [7] used a cell-based assay with mixtures of galectins and exploited the lack of strong binding of the truncated Gal-3 CRD. The presence of unlabeled Gal-1 or -7 increased cell surface association of labeled Gal-3 CRD, an increase that depends on binding of the unlabeled lectin. Co-incubation of labeled Gal-3 CRD with a label-free Gal-1 mutant was inhibited from binding lactose by an E71Q mutant and thus failed to enhance the signal.

Puled-field gradient (PFG) NMR experiments have demonstrated the formation of galectin heterodimers, as opposed to higher order oligomers [7]. For the ^15^N-labeled Gal-3 CRD, the diffusion coefficient, *D*, was 1.39 × 10^−6^ cm^2^/s, a value that is very close to that expected for a monomeric galectin CRD. However, when ^15^N-labeled Gal-3 CRD was paired with unlabeled Gal-7, the ^15^N-PFG derived *D* value decreased to 0.98 × 10^−6^ cm^2^/s, a value close to that observed for a prototype galectin homodimer [97]. Thus, the monomeric Gal-3 CRD was shown to interact with Gal-7 CRD to form a Gal-3:Gal-7 heterodimer. HSQC NMR was also used to validate their physical interaction and to provide insight as to how they interact. By mixing unlabeled Gal-7 with ^15^N-labeled Gal-3, and vice-versa, Miller et al. [7] used HSQC NMR experiments to show that a subset of residues in both Gal-3 CRD and Gal-7 were spectrally perturbed, thus allowing for identification of residues at the heterodimer interface of each galectin. This in turn allowed a structural model of the Gal-3/Gal-7 heterodimer to be developed [7]. The most perturbed residues are within the S-face β-sheet and inter-connecting loops. Using this information, key residues from interacting surfaces between Gal-3 and Gal-7 identified in these HSQC studies, were manually aligned or docked, and molecular dynamics (MD) simulations and energy minimization were performed to optimize the resulting heterodimer structure shown in Figure 4.

Similarly to previous findings with chemokine heterodimers, NMR spectral effects depend on the protein concentration, with effects being more clearly observed when galectin concentrations were kept relatively low to promote some dissociation of their homodimeric states [7]. When Gal-1 CRDs are not strongly associated, as with Gal-1 [8S], hybrid interactions with Gal-7 or the truncated Gal-3 CRD occur more readily than with wild type Gal-1. Furthermore, tandem-repeat Gal-8, with its two monomeric CRDs, also readily associates with resin-bound Gal-7, unlike wild type Gal-1. In toto, these NMR data demonstrate that mixed galectins indeed can form heterodimers and not higher order oligomeric species [7].

### 3.2. Functional Considerations

The concept of galectin-galectin heterodimers presents a new paradigm in the field. At present, there is only one report that has investigated the biological relevance of galectin heterodimers. Hopefully others will follow.

Dings et al. used cell-based assays to demonstrate that combinations of Gal-1, Gal-3 and/or Gal-7 function synergistically in agglutination, cell surface binding, and endothelial cell proliferation and migration [8]. When combined at various molar ratios, Gal-1 and Gal-7 promote the agglutination of red blood cells and leukocytes significantly more than either of these galectins do alone when adjusted for differences in concentration [8]. FACs analysis with FITC-labeled Gal-1 and mouse splenocytes shows that addition of unlabeled Gal-7 enhances binding of FITC-labeled Gal-1 to the surfaces of CD4+ and CD8+ leukocytes, as well as to CD31+ endothelial cells [8]. This can only occur if these two galectins physically associate as hetero-oligomers. FACs analysis also shows that various combinations of Gal-1, Gal-7, and truncated Gal-3 CRD induce apoptosis in these mouse splenocyte cells [8]. Furthermore, Gal-1 and Gal-7 have an effect on mouse 2H11 endothelial cell proliferation and migration. In this regard, various combinations of Gal-1 and Gal-7 attenuate endothelial cell growth compared to either galectin alone. In the wound healing assay used to assess cell migration, combinations of Gal-1 and Gal-7 reduce the extent of endothelial cell migration considerably more than either galectin alone [8]. These effects occur synergistically, and likely would not occur unless pairs of galectins physically associated as heterodimers.

The generation of galectin CRD heterodimers can offer a way to make new functional species in distinct microenvironments, as in protease-rich inflamed tissues that turn full-length Gal-3 into truncated Gal-3 CRD. Non-covalently linked heterodimers are likely to form in vivo under certain circumstances and thereby promote differential functions, representing a physiologic way to achieve further versatility.

## 4. Chemokine–Galectin Heterodimers

Cell migration and activation are mediated by both chemokines and galectins, players that are crucial to these physiological events. The expression of galectins and chemokines is correlated by a galectin-dependent enhancement of chemokine production and secretion, a phenomenon that is found in various cell types, e.g., monocytes [147] or in activated pancreatic stellate cells (that secreted the chemokine CXCL12 promoting pancreatic cancer cell invasiveness) [148,149], rheumatoid synovial fibroblasts [150], osteoarthritic chondrocytes [151,152], endothelial cells [153,154], and bone marrow-derived dendritic cells [155]. Collectively, these functional similarities, along with structural data on chemokine–chemokine and galectin–galectin heterodimers, provided the incentive to validate the fundamental hypothesis that galectins and chemokines interact cross-family to form heterodimers. The first step was to establish an “interactome” of chemokines that associate with Gal-1 and Gal-3. With NMR resonance assignments and structural characterizations in hand for both galectins [156,157] and for the chemokine CXCL12 [158,159], the proof-of-principle was solidly established up to the level of mapping contact sites [9]. This in turn opened the way to explore the potential for this type of interaction and to elicit functional effects in vitro and in vivo, with the perspective of a full-scale network analysis.

Gal-1 and -3, similarly to some chemokines, are upregulated during inflammation and are among the most abundant and best studied galectins in inflammatory diseases. Under inflammatory conditions, metalloproteinases cleave the N-terminal region of Gal-3, resulting in free Gal-3 CRD [160,161,162]. Therefore, the encounter of these galectins with chemokines becomes highly likely and focuses interest on demonstrating heterodimerization in vitro. Solid-phase and surface plasmon resonance-based assays showed that Gal-1 and Gal-3 (as well as the Gal-3 CRD) bind to specific chemokines from CC-, CXC-, XC- and CX3C-families [9]. Surface plasmon resonance was used to show that K_D_ values for CXCL12 binding to Gal-3 and Gal-3 CRD are 80 nM and 34 nM, respectively. Chemokine-galectin binding showed remarkable selectivity, and the analysis of HSQC spectra indicated that the binding epitope on Gal-1 or Gal-3 does not involve the canonical sugar-binding site on the CRD S-face [9], and moreover, allowed us to model the galectin–chemokine heterodimer, a model validated by mutagenesis studies [9].

Figure 5 shows an in silico-generated model of the CXCL12-Gal-3 CRD heterodimer. The largest Gal-3 contact region with CXCL12 involves Gal-3 CRD β-strands β7–β9 (colored orange) on the CRD F-face that opposes the canonical sugar-binding S-face of the CRD β-sandwich (colored green, labeled β4–β6). Even though the carbohydrate binding site on Gal-3 is not directly involved in the interaction with CXCL12, it is indirectly affected in that some sugar binding residues, such as N160, are perturbed enough that carbohydrate binding is influenced. This finding has implications for Gal-3 activity, in that Gal-3 can still bind β-galactosides when CXCL12 is bound. In terms of CXCL12, some residues within its β-sheet (β1–β3) and C-terminal helix interact with the Gal-3 CRD. CXCL12 β1-strand is involved in GAG binding, and CXCL12 binding to heparin attenuates CXCL12-Gal-3 heterodimerization, a finding that has implications for GAG-regulated biology of CXCL12 [9]. In principle, a chemokine dimer can be presented by a GAG and simultaneously to its receptor [163]. A prerequisite for Gal-3-CXCL12 heterodimer formation in vivo would likely be that Gal-3 binds as well to GAGs, and indeed there are reports that it does [112,164], an observation requiring that the dimer interface does not overlap with the GAG/receptor binding site.

Gal-3 is expressed by tumor-associated macrophages that are responsible for reducing the mobility of other leukocyte subsets, especially CD8+ T cells [165]. In this respect, the binding of Gal-3 to chemokines appears to be mechanistically relevant [9,166]. Indirectly, Gal-3 inhibits chemokine-dependent migration of CXCR3-expressing lymphocytes by preventing diffusion of IFNγ and in turn the upregulation of chemokines CXCL9, CXCL10, and CXCL11. Recently, evidence for direct biological effects from Gal-3–chemokine heterodimers has been reported with the inhibition of CXCL12- or CCL26-mediated chemotaxis (CXCL12-induced T-cell and neutrophil migration and CCL26-induced eosinophil migration) being attenuated by heterodimer formation with full-length Gal-3 or its CRD [9].

Gal-3/CXCL12 heterodimers have been detected on the cell surface, and a model of the Gal-3/CXCL12 heterodimer bound to CXCR4 within the membrane shows that Gal-3 binds CXCL12 with no steric hindrance to CXCL12 in binding its receptor CXCR4 [167]. This model stands somewhat in contrast to the X-ray crystal structure of viral chemokine vMIPII bound to CXCR4, which shows that it is the chemokine monomer, and not the homodimer, that binds to CXCR4 [65]. Whereas chemokine ligand CXCL12 indeed binds receptor CXCR4 as a monomer, it is the CXCL12/Gal-3 heterodimer that is modeled to bind to CXCR4 via direct interactions with CXCL12 and not with Gal-3 CRD [167]. These CXCL12/Gal-3 complexes occur at concentrations that are lower than the actual K_D_ values would predict. The reason for this is likely that cell surface structures and molecules, such as GAGs and the glycocalyx, facilitate heteromerization. This model implies that Gal-3 can influence CXCL12 signaling by forming a ternary complex with CXCR4 that induces a conformational change in the GPCR [167]. With chemokine heterodimers inducing chemokine receptor heterodimerization, it is possible that a galectin–chemokine heterodimer can bring together a pair of chemokine and galectin receptors. Although speculative, the formation of a chemokine–galectin heterodimer could have wide-ranging biological consequences, opening a new, broad area for study.

## 5. Conclusions

As small effector molecules, chemokines and galectins are crucial to various biological functions. Additionally, although they can function within their individual families as monomers or homodimers, it is now evident that they can also cross-interact within each of their respective families, as well as between family members as chemokine-galectin heterodimers. The formation of these novel heterodimers has been shown to be functionally significant, to differentiate and expand on their known activities, and to fine-tune those activities, especially when both are present in the same cellular environment. Some of these are summarized in Table 1.

These reports on chemokine and galectin heterodimers have increased our understanding of the functional versatility and cooperation among galectins and chemokines. Moreover, due to the growing interest in this area, a number of labs have initiated their own investigations on these and other small protein effector molecules. The heterodimer paradigm among these small effector molecules is compounded by several observations, namely some chemokine and galectin concentrations are simultaneously elevated at some cellular/tissue sites, e.g., inflammation; both chemokines and galectins can interact with each other within and between their families in diverse manners, and while galectins engage with cell surface glycans, chemokines engage with cell-surface GAGs. Overall, this provides direction to additional structural and functional studies of chemokine and galectin heterodimers, along with production of stable heterodimers that will enhance the physiological significance of these new classes of molecular hybrids and possibly their use as pharmaceutical drugs in clinical practice. However, when new paradigms are initially reported, such as galectin and chemokine heterodimers, there is always some controversy and level of unacceptance. This is particularly true in terms of biological relevance with the new paradigm. Nevertheless, skepticism is a sound scientific practice that then prompts investigators to delve into the new paradigm and experiment further to prove or disprove it.

## Figures and Tables

**Figure 1 ijms-24-14083-f001:**
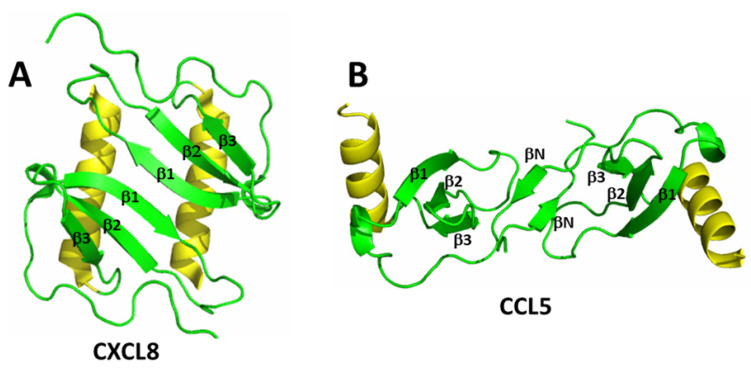
(**A**) CXC chemokine CXCL8 homodimer (Interleukin-8, PDB access code 1IL8) and (**B**) CC chemokine CCL5 homodimer (RANTES, PDB access code 5COY) are shown. C-terminal helices are colored yellow, and the remainder of the sequences are colored green. The illustrations of structures shown in this figure were made by Dr. Michelle C. Miller.

**Figure 2 ijms-24-14083-f002:**
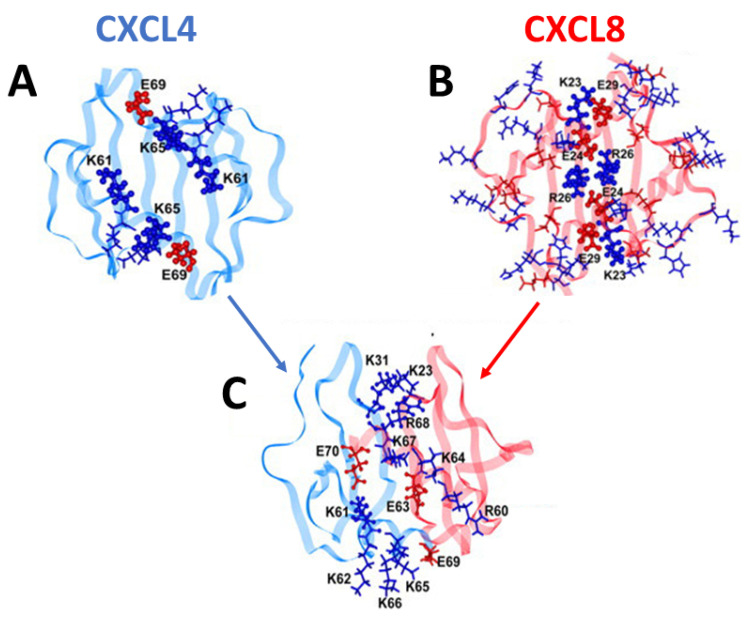
Modeled structures of the CXCL4/CXCL8 heterodimer based on chemical shift perturbations in HSQC spectra and in silico molecular mechanics and dynamics simulations. The CXCL4 (platelet factor 4) homodimer is shown in blue (**A**), and the CXCL8 (interleukin-8) homodimer is shown in red (**B**), with the heterodimer formed from a monomer subunit from each shown in (**C**). Positively charged lysine and arginine residues are shown in dark blue as labeled, and negatively charged glutamate residues are shown in dark red as labeled. Electrostatic interactions among these residues in particular stabilize both homodimers and heterodimers. The illustrations of structures shown here were made by Dr. Irina Nesmelova.

**Figure 3 ijms-24-14083-f003:**
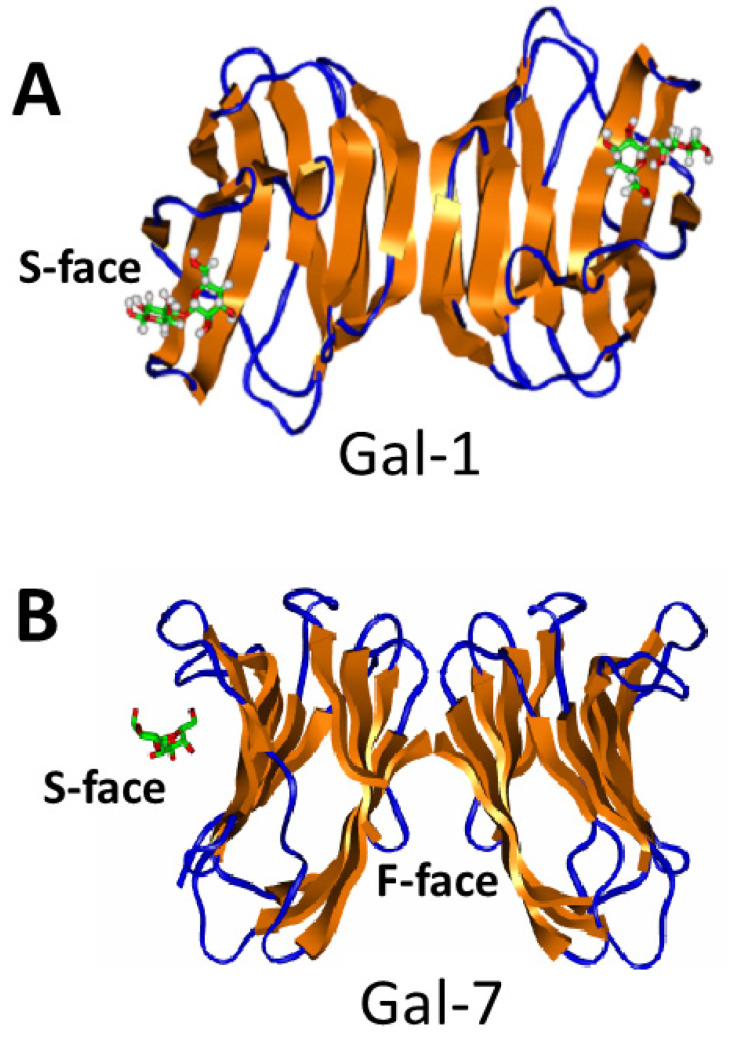
Galectin homodimers. Two different types of prototype galectin homodimers are illustrated. (**A**) Gal-1 homodimer (PDB lGZW). (**B**) Gal-7 homodimer (PDB lBKZ). The sugar binding S-face and opposing F-face are indicated in (**B**), and molecules of bound lactose are shown as ball-and-stick structures in both (**A**,**B**).

**Figure 4 ijms-24-14083-f004:**
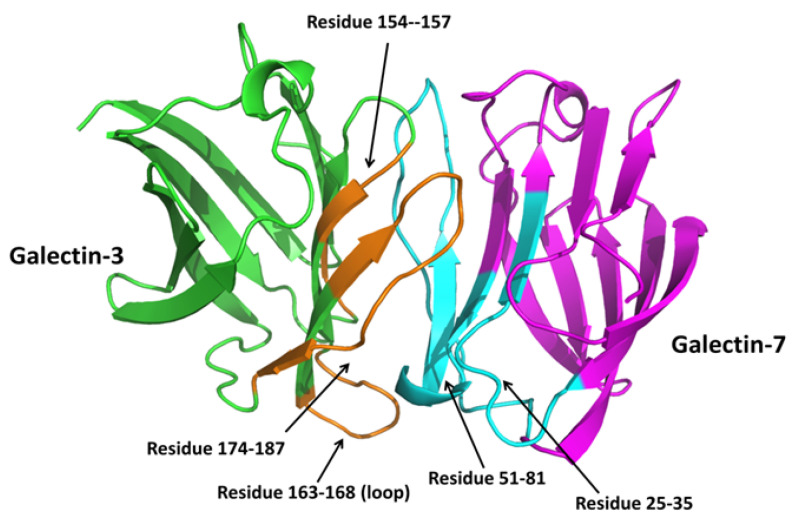
Model of the Gal-3/Gal-7 heterodimer. Using HSQC-derived information, in silico studies were performed by manually aligning and docking key residues from the Gal-3 CRD with those from Gal-7 to form the heterodimer, followed by MD simulations and energy minimization to thermodynamically optimize interactions. This figure has been re-printed from Miller et al. [7] with permission of the publisher.

**Figure 5 ijms-24-14083-f005:**
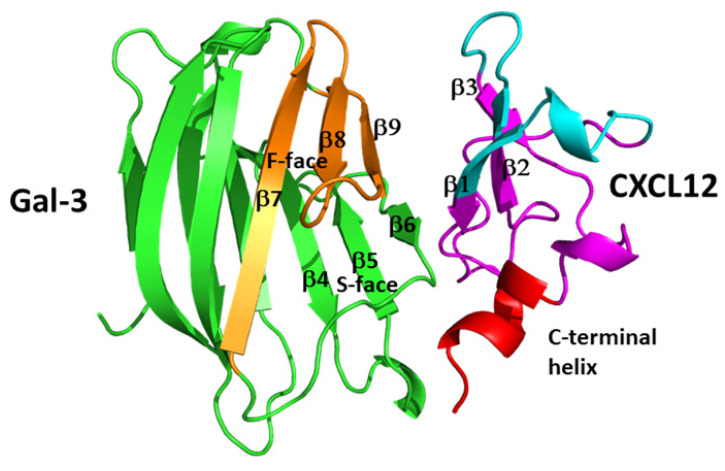
Modeled structure of the CXCL12 and Gal-3 CRD heterodimer based on NMR chemical shift and intensity changes from HSQC experiments with CXCL12 and Gal-3 CRD. Analyses of the HSQC spectra were used to manually dock the two species and energy minimization the complex using MD simulations as discussed in [6]. This modeled structure was produced by Dr. Kanin Wichapong.

**Table 1 ijms-24-14083-t001:** Some heterodimers with functional consequences.

Heterodimer	Functional Effects	References
CXCL4–CXCL8	Enhance EC proliferation, Baf3 cell migration	[3]
CCL3/4, CCL2/8	Activate CCR5 intra-cellular signaling	[1]
CXCL4–CCL5	Modulate GPCR-mediated signal transduction	[4,5,6]
CXCL4L1-MIF	Regulate inflammation and thrombus formation	[55]
Gal-7-Gal-1 or Gal-3	Increase EC and leukocyte cell surface binding	[7]
	Enhance hemagglutination and leukocyte apoptosis	
	Attenuate EC growth and migration	
Gal-3-CXCL12	Inhibit CXCR3-expressing lymphocyte migration	[9,166]
	Enhance upregulation of CXCL9, L10, L11	
	Inhibit CXCL12-induced T-cell, neutrophil migration	
Gal-3-CCL26	Inhibit CCL26-induced eosinophil migration	[9]

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
