# Peer review of "Heterologous Interactions with Galectins and Chemokines and Their Functional Consequences"

_ijms, 2023, doi:10.3390/ijms241814083_

Round 1

Reviewer 1 Report

The review paper “Heterologous interactions with galectins and chemokines and their functional consequences” presents a comprehensive summary of the interesting paradigm highlighting an importance of inter-molecular interactions and heterocomplexes of galectins and cytokines in regulating multiple cellular responses. This concept is very attractive and provides good ideas for future studies. I recommend acceptance with a few minor edits/questions to be addressed by the author:

1) Greek font: several places need to replace ‘b’ to ‘beta’, e.g. lines 60, 282, 357, 439.

2) line 175: delete (2022)

3) lines 274-275: How much do we know about galectin-16? The statement about galectin-16 relates to the recombinant galectin, which should be indicated.

4) line 394: the statement “…galectins serve as bridges between cells…” needs support with additional references which confirm direct galectin-induced cell aggregation (e.g. PMID: 12962152).

Author Response

We have made all the changes requested by Reviewer 1:

1) Greek letters added.

2) (2022) deleted on line 175

3) added new lines 278-282 to address more about what is known about Gal-13, -14 and -16 

4) added reference PMID 12962152 

Reviewer 2 Report

Overall, the manuscript "Heterologous interactions with galectins and chemokines and their functional consequences" makes a favorable impression, the review is well illustrated (5 figures of three-dimensional structures of heterologous complexes are given), and heterologous complexes between galectins and chemokines are described. Given the diversity of chemokines and galectins, this review would look even more favorable if there were a table with columns with interaction partners and functional consequences of the formation of heterologous complexes between them, as well as references to solved three-dimensional structures (if available).   

Author Response

Table 1 was added within the Conclusions section to list some heterodimers with functional consequences as requested.

Unfortunately, there are no empirically determined structures known for any of these heterodimers. At present, only NMR-based in silico-generated models have been reported. 

Reviewer 3 Report

The title heterologous interactions...and their functional consequences implies that also biophysics of molecular interaction are in the foreground. In the main text there are more phenomenological descriptions of the molecules and possible heterodimers. The authors should stress more the physiological important molecular interactions, like electrostatic interactions, molecular mechanics, binding cooperatively and in sum the possible outcome of such molecular machinery. Furthermore, a statistical assessment of the cooperation between members of the galectine and chemokine families and its physiological significance would help.

Minor correction: line 123: chemokine:GAG...check spacing.

Author Response

Even though Reviewer 3 raises some good points in terms of identifying physiologically important interactions, binding cooperativity, and out come of this molecular machinery, this seems to be beyond the scope of this review. We are currently writing a new manuscript based on a mutagenesis investigation of such important interactions based on the models currently in hand. However, no empirically determined structures have yet been reported on any of these heterodimers. 

Unless I misunderstood Reviewer 3's request, a statistical assessment of cooperation between members of galectins and chemokines also seems like a new study would need to be performed to address this issue, again beyond the scope of this review. 

line 123 has been changed from "chemokine:GAG" to "chemokine-to-GAG" 

Round 2

Reviewer 3 Report

The reply of the authors has convinced - it should be, indeed, the subject of a new paper because such biophysical mechanism of interaction come more and more into focus of cell- and molecular biology.

Author Response

I thank this reviewer for his comments, and we will certainly continue in this line of research. However, I did not see any issues that require my further modifications to this manuscript. Thank you.